# Two Year Neurodevelopmental Outcome after Fetoscopic Laser Therapy for Twin–Twin Transfusion Syndrome: Comparison with Uncomplicated Monochorionic Diamniotic Twins

**DOI:** 10.3390/children10071250

**Published:** 2023-07-20

**Authors:** Ángel Chimenea, Lutgardo García-Díaz, Guillermo Antiñolo

**Affiliations:** 1Department of Materno-Fetal Medicine, Genetics and Reproduction, Institute of Biomedicine of Seville (IBIS), Hospital Universitario Virgen del Rocio, CSIC, University of Seville, ES-41013 Seville, Spain; angel.chimenea@hotmail.com (Á.C.); gantinolo@us.es (G.A.); 2Fetal, IVF and Reproduction Simulation Training Centre (FIRST), ES-41010 Seville, Spain; 3Centre for Biomedical Network Research on Rare Diseases (CIBERER), ES-41013 Seville, Spain

**Keywords:** fetoscopic laser therapy, neurodevelopmental outcome, monochorionic twin, twin–twin transfusion syndrome, cerebral palsy

## Abstract

Background: Twin–twin Transfusion Syndrome (TTTS) represents a significant complication in monochorionic twin pregnancies, caused by an unbalanced shunting of blood through intertwin placental vascular anastomoses. Despite advances in fetoscopic laser surgery, TTTS is still associated with a high rate of cerebral injury. However, there are no studies comparing these pregnancies with uncomplicated monochorionic diamniotic (MCDA) twin pregnancies, establishing the baseline risk of neurodevelopmental impairment. The aim of this study is to evaluate the odds of neurodevelopmental impairment in MCDA twins who undergo fetoscopic laser surgery for twin–twin transfusion syndrome, in comparison to a cohort of uncomplicated MCDA twin pregnancies. Study design: This is a retrospective cohort study of children born from MCDA twin pregnancies at a single center between 2008 and 2019. A routine, standardized follow-up assessment was conducted at a minimum of 2 years after delivery. The primary outcome of this was a 2 year neurodevelopmental impairment. Neurological, motor, and cognitive development was assessed by using the revised Brunet–Lézine scale. Results: 176 children met the enrolment criteria. Of these, 42 (24%; TTTS group) underwent fetoscopic laser surgery for TTTS during pregnancy, and 134 (76%; uncomplicated MCDA group) were uncomplicated MCDA pregnancies. The primary outcome was found in four children (9.52%) in the TTTS group and ten children (7.46%) in the uncomplicated MCDA group (*p* = 0.67, aOR 2.82, 95% CI 0.49–16.23). Major neurologic impairment was found in 2.38% after fetoscopic laser surgery and 1.49% in uncomplicated MCDA twins (*p* = 0.70, aOR 0.97, 95% CI 0.22–4.24). The data were adjusted by birth order, birth weight, and gestational age at birth. Conclusions: The outcome in MCDA twins who underwent fetoscopic laser surgery for TTTS is comparable to the outcome in uncomplicated MCDA twins. Our findings emphasize the need for long-term neurodevelopmental follow-ups in all children from monochorionic twin gestations.

## 1. Introduction

Twin–twin transfusion syndrome (TTTS) is a significant complication observed in monochorionic (MC) twin pregnancies, occurring as a serious complication in 10% to 20% of those gestations [1]. TTTS occurs as a result of an asymmetrical redistribution of blood through intertwin placental vascular anastomoses, leading to a discrepancy in blood volume distribution, resulting in a blood donor fetus and a recipient fetus. Without treatment, this disease has a mortality rate of close to 90% for both twins and a considerable risk of long-term complications for surviving fetuses, including neurodevelopmental impairment (NDI) and cerebral palsy (CP) [2,3].

As described over two decades ago, fetoscopic laser coagulation of the anastomoses is widely regarded as the standard treatment for TTTS, and its superiority over serial amnioreduction has been proven [1,4,5,6,7,8,9]. This endoscopic therapy may achieve a survival rate of close to 90% for at least one fetus and 40–50% for both twins, finding good results even in advanced Quintero stages [10].

There are actually two techniques for performing laser fetoscopy [11]. Traditionally, the selective coagulation of anastomoses (the selective technique) has been employed, wherein only the visible anastomoses between the fetuses are coagulated. In the last decade, a technique based on complete dichorionization (the Solomon technique) has been developed. This involves performing selective coagulation and subsequently coagulating the placental equator, connecting the areas where selective coagulation was previously carried out. The Solomon technique has shown to increase survival rates while reducing the recurrence of Twin–Twin Transfusion Syndrome (TTTS). However, it is associated with an increased risk of placental abruption and a lower gestational age at birth [12].

However, there are concerns related to its possible impact on fetal neurodevelopment [13]. Despite advances in the technique, improved outcomes in terms of morbidity and mortality, and increased operator experiences, laser surgery as a treatment for TTTS has been associated with varying rates of severe cerebral injury (3% to 16%) [8,14,15,16] and neurodevelopmental impairment (8% to 18%) [17].

In spite of the growing evidence regarding the occurrence of NDI after laser fetoscopy, there are no studies that compare it with uncomplicated MCDA twin pregnancies. These results may help to have a clear appreciation of these differences and the possible impact of this intervention.

This study aims to analyze the occurrence and frequency of the 2 year NDI rate after laser fetoscopy during gestation due to TTTS, compared to the neurodevelopment in uncomplicated monochorionic diamniotic (MCDA) twin gestations.

## 2. Materials and Methods

### 2.1. Study Design, Setting, and Population

This retrospective cohort study was conducted at the Virgen del Rocío University Hospital of Seville, between 2008 and 2019. The department of Maternal-Fetal Medicine, Genetics, and Reproduction at the Virgen del Rocío University Hospital has served as a referral unit for the laser treatment of TTTS pregnancies in Andalucía (Spain) since 2008. In this study, we aimed to compare the neurodevelopmental outcomes at 2 years of age between fetuses undergoing laser fetoscopy for TTTS at our center and uncomplicated MCDA twin pregnancies.

The management protocols for MCDA twin gestations were carried out according to those established by the ISUOG, and they were adapted in their respective updates during the study period [18]. Chorionicity and pregnancy dating were determined by a first trimester ultrasound. A detailed mid-trimester ultrasound was performed in all cases at 18.0–21.6 weeks.

The device utilized for first trimester and mid-trimester evaluations was the Voluson 730 Expert (GE Medical System, Milwaukee, WI, USA). The monitoring of the monochorionic pregnancies was conducted using the Voluson E8 & E10 (GE Medical System, Milwaukee, WI, USA).

Fetoscopic laser procedures were performed using the selective coagulation technique throughout the study period. Our group transitioned to adopting the Solomon technique, starting in the year 2020. The fetoscopic procedures were conducted by two specialists in Fetal Medicine, since their establishment at the center in 2008. The photocoagulation procedures were conducted utilizing a multidiode laser fiber with a diameter of 600 μm. The laser system employed delivered pulses of 50 W energy, ensuring precise and effective photocoagulation during the procedure.

Neurodevelopmental assessment was routinely assessed in all the uncomplicated MC twin newborns, as well as the surviving children after intrauterine laser therapy.

### 2.2. Outcomes

The primary outcome of this study was the neurodevelopmental status at 2 years of corrected age, which was evaluated using the revised Brunet–Lézine scale. The revised Brunet–Lézine psychomotor development scale has its origin in the scale published by these authors in 1951, inspired by the Gessel scale [19].

Items were distributed around four areas: (1) posture: postural control or general motor skills; (2) coordination: visual–motor coordination; (3) language: comprehension and expression; and (4) sociability: social relations. The evaluation was structured in three visits (6, 12, and 24 months), evaluating 10 items in each visit (6 experimental items and 4 that were established through the parent’s interview), evaluating a total of 30 items. The score for each item was dichotomous (0/1), depending on whether it had been reached or not. Finally, a developmental age was obtained and a global developmental quotient (DQ), the result of dividing the real age by the developmental age.

Infants were classified as having NDI if their DQ on the revised Brunet–Lézine scale was below 85, or if they exhibited level 3 CP. Severe NDI was further categorized as infants with a DQ below 60. These thresholds are standardized and have already been used by other groups in the field of obstetrics and perinatology [20,21].

### 2.3. Exposures

In our study, infants born from gestations in which laser fetoscopy was performed after the diagnosis of TTTS were selected as the exposed group. Fetuses with intrauterine death, a pre-viable stage of delivery, and children with no postnatal follow-up were excluded from this study. The control group constituted children from uncomplicated MCDA twin pregnancies. To control for potential confounding factors, we excluded children from pregnancies with any of the following complications: stillbirth, severe pre-eclampsia, antepartum cardiotocography pathology, severe intrauterine growth restriction, selective intrauterine growth restriction, twin-to-twin transfusion syndrome, twin anemia-polycythemia sequence, congenital anomaly, and a gestational age less than 32 weeks or greater than 38 weeks at the time of delivery, as well as those with uncertainty regarding their gestational age at birth. As in the previous group, infants who could not be followed up completely were excluded.

### 2.4. Data Source

The data from pregnant women and children were extracted from electronic health records and subsequently anonymized before analysis.

### 2.5. Statistical Analysis

Numeric values were reported as mean ± standard deviation, assuming a normal distribution, whereas categorical variables were presented as frequencies and percentages. Univariable comparisons of categorical variables were performed, using either the chi-square test or Fisher’s exact test. Student’s *t*-test was used for comparing normally distributed continuous variables, while the Mann–Whitney U-test was employed for non-normally distributed variables. Binary logistic regression models were utilized to calculate odds ratios (OR) with their corresponding 95% confidence intervals (CI).

To account for significant confounding factors, the OR were adjusted (aOR) for birth order, birth weight (per gram), and gestational age at birth (per day). The statistical analysis was conducted using the SPSS 27.0 software package (SPSS Inc., Chicago, IL, USA). All hypothesis tests were two-sided, and a significance level of *p* < 0.05 was considered statistically significant.

## 3. Results

Considering the enrollment criteria described above, a total of 176 children were finally included in the study. Of these, the exposed group (TTTS group) included 42 children, while the control group (uncomplicated MCDA group) included 134 infants. The flowchart illustrating the participant enrollment can be observed in Figure 1.

After applying the exclusion criteria, the follow-up rate of the ‘2-year examination’ was 97.8% in the uncomplicated MCDA group and 53.8% in the TTTS group (134/174 and 42/78, respectively). This situation is explained by the fact that this hospital serves as a specialized referral center for fetal therapy, and after delivery, the patients were followed up at their hospitals of origin.

Table 1 presents the baseline characteristics of the study participants, and the data relating to laser fetoscopy are summarized in Table 2. We found a significantly lower maternal age in the TTTS group (33 years vs. 41 years, *p* < 0.001), in addition to a higher rate of ART conception (44.8% vs. 20.9%, *p* = 0.02). However, despite the statistical significance, we did not find a causal link in our sample or in the consulted related literature. No other significant differences in maternal and obstetrical characteristics were found between the groups.

Table 3 presents neurodevelopmental status at 6, 12, and 24 months of the corrected age of the children included for follow-up evaluation. The children who underwent laser fetoscopy during pregnancy presented a similar rate of NDI as the children with uncomplicated MCDA gestations, including at 6 months (9.52% vs. 7.46%, *p* = 0.67, aOR 3.70, 95% CI 0.66–20.57), 12 months (9.52% vs. 6.72%, *p* = 0.54, aOR 2.64, 95% CI 0.44–15.70), and 24 months (9.52% vs. 7.46%, *p* = 0.67, aOR 2.82, 95% CI 0.49–16.23).

Considering the rate of severe neurodevelopmental impairment, we also found no statistically significant differences at 6 months (1.49% vs. 0%, *p* > 0.99), 12 months (1.49% vs. 2.38%, *p* = 0.70, aOR 0.97, 95% CI 0.22–4.24), and 24 months (1.49% vs. 2.38%, *p* = 0.70, aOR 0.97, 95% CI 0.22–4.24) between the two cohorts.

## 4. Discussion

### 4.1. Principal Findings

The main objective of this study was to determine the rate of 2 year NDI after laser fetoscopy for the treatment of TTTS, compared with the neurodevelopment of those born from uncomplicated MCDA twin pregnancies. This study confirms that laser fetoscopy after a diagnosis of TTTS achieves comparable results to uncomplicated MCDA twin gestations, in terms of 2 year NDI. The results also show a 2 year NDI rate of 7.46% in the group of uncomplicated MCDA gestations; therefore, it is advisable even for these fetuses to follow up their neurodevelopment at least up to 2 years of life.

### 4.2. Interpretation of Findings and Comparison with Other Published Evidence

The exact pathogenesis of cerebral injury in twin–twin transfusion syndrome (TTTS) is not yet fully understood or clearly defined. Cerebral injury in TTTS is thought to occur at all Quintero stages, and this might derivate from a hemodynamic and hematological imbalance before laser surgery [22,23]. Nevertheless, other factors have an important influence on neurodevelopment, such as postnatal injury associated with prematurity [24] and low birth weight [25]. These factors also appear in uncomplicated MCDA twin gestations; therefore, the effect of this disease and its treatment requires a comparison between the two groups. The absence of a control group has been a recurrent limitation expressed by the authors in previous studies [26].

There have been previous studies reporting neurodevelopmental outcomes after laser surgery for TTTS. However, as far as we know, this is the first study to compare the rate of NDI after laser fetoscopy with a cohort of uncomplicated MCDA gestations, supporting the safety of laser fetoscopy in these terms, as well as establishing the baseline risk of NDI in MCDA twins. Other authors conducted studies with a control group, but based on dichorionic twin gestations [15,27].

There is a recent systematic review analyzing 26 studies performed between 1999 and 2021, which descriptively reports the rate of NDI after laser fetoscopy [28]. The authors included 2443 patients in their review, reporting a rate of CP ranging from 2% to 18% (mean 5%) and an overall rate of NDI that ranged from 4% to 18% (mean 9%). When comparing our data with those coming from this systematic review and other studies [29,30] (data summarized in Figure 2), we can observe a very variable rate of NDI, possibly due to differences in the evaluation scale, the evaluation subjectivity, and the small sample size of some studies. In our case, the severe 2 year NDI rate was 2.38%, close to the lower range in studies in this synthesis. The overall rate of NDI in our series was 9.52%, and this data are in the range of those international studies published on this subject [14,23,26,27,28,29,31,32,33,34,35,36,37,38,39,40,41,42,43,44,45,46,47,48,49,50,51,52,53,54,55].

To ascertain whether these data represent a true increase related to the disease, we performed a comparison with a cohort of children born from uncomplicated MCDA gestations, adjusting the results for weight and gestational age, as these are considered to be the main factors related to neurodevelopment [24,25,56]. In our study, the neurodevelopmental outcome of the TTTS group did not differ statistically from that found in the control group of uncomplicated MCDA gestations, despite the greater hemodynamic insult in the exposed group. This suggests that laser fetoscopy may have a beneficial effect on these gestations.

This beneficial effect has also been evidenced in a recent meta-analysis conducted by Yan et al.’s group [57]. The authors concluded that after laser fetoscopy, the neurodevelopmental outcomes in terms of NDI become comparable between MCDA twin gestations affected by TTTS and dichorionic gestations (odds ratio [OR], 1.71; 95% confidence interval [CI], 0.81–3.63, I^2^ = 0%). They also found no significant differences in the occurrence of cerebral palsy (Peto odds ratio [OR], 2.00; 95% CI, 0.88–4.52, I^2^ = 0%).

Our data regarding severe neurological impairment in uncomplicated MCDA gestations (1.49%) are similar to that recently reported in a secondary analysis of the EPIPAGE2 study [58]. In this study, out of a total of 926 evaluated children, 228 were from MCDA twin gestations. Among them, neurobehavioral impairment was assessed in 143 cases, revealing a rate of moderate/severe cerebral palsy of 0.9% in MCDA gestations (compared to the 1.5% in dichorionic gestations used as a control group, *p* = 0.46). Since the neurodevelopmental outcome measures did not include children who experienced TTTS during pregnancy, we were unable to make a direct comparison with the TTTS group in our study.

### 4.3. Clinical Implications

Our results have shown that after laser fetoscopy, there is a low percentage of NDI and cerebral palsy at two years of age. However, an unexpected rate of NDI in uncomplicated MCDA gestations was evidenced. These findings are in line with those reported by our group in previous studies [55], emphasizing the importance of a neurodevelopment postnatal follow-up in uncomplicated MCDA twin gestations. This conclusion has also been reached by other fetal therapy groups around the world [59].

### 4.4. Research Implications

Considering the variability in existing clinical practice, we recommend replicating this study in other populations, incorporating data from different fetal surgery centers. The impact of TTTS may differ in relation to the Quintero stage at diagnosis; therefore, future studies with adequate sample sizes are needed to assess the impact of early or advanced stages of the disease. Further studies are needed to investigate neurodevelopment until scholar age.

### 4.5. Strengths and Limitations

The presence of a control group consisting of uncomplicated MCDA twin gestations represents a major strength of this study. This control group allows for the establishment of a baseline risk against which the TTTS gestations can be compared, enhancing the internal and external validity of this study.

Due to the retrospective nature of the study design, the inclusion of patients was based on specific criteria within a defined timeframe. Another main limitation is the limited follow-up period, as the evaluation of the survivors was conducted only until 2 years of corrected age. Neurodevelopmental assessments performed at this early stage may only partially predict potential alterations that could manifest at later ages. It is essential to acknowledge that neurodevelopment is a dynamic process, and significant developmental differences may emerge as children grow older.

Comparing our results with those of other studies posed challenges, due to the use of different psychomotor scales. The choice of specific psychomotor scales may be influenced by various factors, such as cultural contexts, research objectives, and available resources. This variability in assessment tools makes it difficult to establish direct comparisons and generalize the findings across different studies. Furthermore, it is important to acknowledge that our study did not account for potential confounding factors related to social and environmental influences on neurodevelopment. Future studies should consider incorporating comprehensive data collection and stratification for these factors to provide a more nuanced understanding of their contributions to neurodevelopment.

## 5. Conclusions

In this present study, the rate of 2 year NDI and severe NDI in MCDA twins treated with laser fetoscopy was found to be comparable to the outcome in uncomplicated MCDA twins. These results support the long-term safety of laser fetoscopy in TTTS treatment. Our findings emphasize the need for long-term neurodevelopmental follow-up in all children from MCDA twin gestations.

## Figures and Tables

**Figure 1 children-10-01250-f001:**
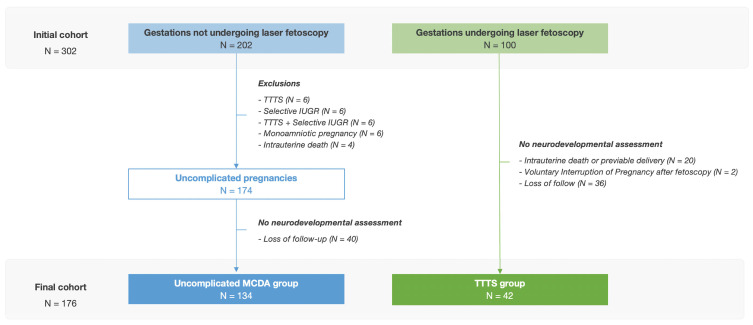
Flow chart of participant enrolment.

**Figure 2 children-10-01250-f002:**
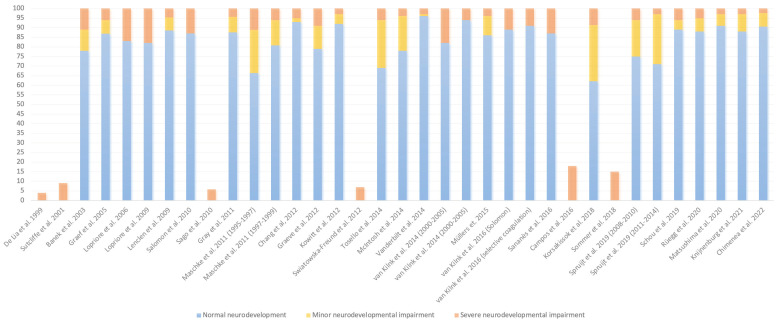
Rate of neurodevelopmental impairment after laser fetoscopy in monochorionic twin pregnancies complicated with Twin–Twin Transfusion Syndrome. Comparative analysis [14,23,26,27,28,29,31,32,33,34,35,36,37,38,39,40,41,42,43,44,45,46,47,48,49,50,51,52,53,54,55].

**Table 1 children-10-01250-t001:** Baseline characteristics.

Variable	Uncomplicated MCDA Group n = 134 (76.1%)	TTTS Group n = 42 (23.9%)	*p* Value
Maternal age, years	41 ± 5.4	33 ± 6.5	<0.001
Maternal weight, kilograms	64.2 ± 14.7	67.6 ± 13.9	0.29
Body Mass Index, kilograms/meters	23.8 ± 6.1	25.4 ± 4.8	0.23
Previous CS, n (%)	22 (16.4)	10 (24.1)	0.37
Previous vaginal birth, n (%)	88 (65.7)	23 (55.2)	0.33
ART conception, n (%)	28 (20.9)	19 (44.8)	0.02
Pregestational diabetes, n (%)	0	1 (3.4)	0.30
Smoking habit, n (%)	32 (23.9)	10 (24.1)	0.98

ART: Assisted Reproductive Technology; CS: Cesarean Section.

**Table 2 children-10-01250-t002:** TTTS group data and outcomes.

Variable	TTTS Group n = 30 MCDA Pregnancies/42 Survivors after Laser Fetoscopy
Location of the placenta (posterior)	63.3% (19/30)
GA at laser fetoscopy, weeks	21.1 (1.6)
TTTS stage–median (range)	2 (1–3)
Stage I	33.3% (10/30)
Stage II	43.4% (13/30)
Stage III	23.3% (7/30)
Stage IV	0
Survival after laser fetoscopy	
At least one survivor	100% (30/30)
Two survivors	53.3% (16/30)
Surviving fetus (donor)	58.8% (23/42)
GA at birth, weeks	31.3 (3.3)
Birth weight (grams)	1437 (565.5)
Sex (female)	59.5% (25/42)
Apgar 5 min < 7	11.9% (5/42)

Data are presented as n (%), mean ± standard deviation or median (range). GA: Gestational Age; TTTS: Twin–Twin Transfusion Syndrome.

**Table 3 children-10-01250-t003:** Long-term neurological morbidity in twin–twin transfusion syndrome treated with fetoscopic laser surgery vs. uncomplicated monochorionic diamniotic twins.

Neurodevelopmental Status	Uncomplicated MCDA Group n = 134 (76.1%)	TTTS Group n = 42 (23.9%)	*p* Value	OR (95% CI)	aOR (95% CI) **
6 months					
Normal neurodevelopment	92.54% (124/134)	90.48% (38/42)	Ref.	Ref.	Ref.
Neurodevelopmental impairment	7.46% (10/134)	9.52% (4/42)	0.67	1.31 (0.39–4.40)	3.70 (0.66–20.57)
Severe neurological impairment	1.49% (2/134)	0% (0/142)	>0.99	*	*
12 months					
Normal neurodevelopment	93.28% (125/134)	90.48% (38/42)	Ref.	Ref.	Ref.
Neurodevelopmental impairment	6.72% (9/34)	9.52% (4/42)	0.54	1.46 (0.43–5.01)	2.64 (0.44–15.70)
Severe neurological impairment	1.49% (2/134)	2.38% (1/42)	0.70	1.61 (0.14–18.21)	0.97 (0.22–4.24)
24 months					
Normal neurodevelopment	92.54% (124/134)	90.48% (38/42)	Ref.	Ref.	Ref.
Neurodevelopmental impairment	7.46% (10/134)	9.52% (4/42)	0.67	1.31 (0.39–4.40)	2.82 (0.49–16.23)
Severe neurological impairment	1.49% (2/134)	2.38% (1/42)	0.70	1.61 (0.14–18.21)	0.97 (0.22–4.24)

* OR cannot be calculated reliably due to zero events in at least one group. ** Adjusted for birth order, birth weight (per gram), and gestational age at birth (per day).

## Data Availability

The data that support the findings of this study are available from the corresponding author, [G.A.], upon reasonable request.

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
