# Peer review of "Two Year Neurodevelopmental Outcome after Fetoscopic Laser Therapy for Twin–Twin Transfusion Syndrome: Comparison with Uncomplicated Monochorionic Diamniotic Twins"

_children, 2023, doi:10.3390/children10071250_

Round 1
Reviewer 1 Report
The research is very interesting and well conducted. The following are some critical points.
DISCUSSION
· Authors are requested to discuss the possible role of maternal age difference between the two groups
· Authors are asked to discuss two important recent publications on the topic:
o Hoarau D, Tosello B, Blanc J, Lorthe E, Foix-L'Helias L, D'Ercole C, Winer N, Subtil D, Goffinet F, Kayem G, Resseguier N, Gire C; EPIPAGE2 Obstetric Writing Group*. Chorionicity and neurodevelopmental outcomes at 5½ years among twins born preterm: the EPIPAGE2 cohort study. BJOG. 2023 Apr 17. doi: 10.1111/1471-0528.17460
o Yan S, Wang Y, Chen Z, Zhang F. Chorionicity and neurodevelopmental outcomes in twin pregnancy: a systematic review and meta-analysis. J Perinatol. 2023 Feb;43(2):133-146. doi: 10.1038/s41372-022-01534-y
o Giannubilo SR, Fiorelli A, Marzioni D, Tossetta G, Capogrosso G, Ciavattini A. Maternal Inherited Thrombophilia in Monochorionic Twin Pregnancy with Twin-Twin Transfusion Syndrome. J Clin Med. 2022 Nov 29;11(23):7054. doi: 10.3390/jcm11237054
REFERENCES
Authors are requested to remove the double numbering of the list of references
Author Response
- The research is very interesting and well conducted. The following are some critical points.
- We would like to begin our response by expressing our gratitude for the reviewer's evaluation and valuable contributions to the publication, which we find highly interesting.
DISCUSSION
- Authors are requested to discuss the possible role of maternal age difference between the two groups
- We have conducted an extensive literature review and have found no references supporting a statistically significant association between maternal age and the risk of Twin-to-Twin Transfusion Syndrome (TTTS). Moreover, our sample data does not provide evidence for causality, leading us to conclude that no causal relationship exists. Although statistically significant, these differences are not clinically relevant. We have included this information in the revised manuscript (lines 150 to 151).
- Authors are asked to discuss two important recent publications on the topic:
Hoarau D, Tosello B, Blanc J, Lorthe E, Foix-L'Helias L, D'Ercole C, Winer N, Subtil D, Goffinet F, Kayem G, Resseguier N, Gire C; EPIPAGE2 Obstetric Writing Group*. Chorionicity and neurodevelopmental outcomes at 5½ years among twins born preterm: the EPIPAGE2 cohort study. BJOG. 2023 Apr 17. doi: 10.1111/1471-0528.17460
Yan S, Wang Y, Chen Z, Zhang F. Chorionicity and neurodevelopmental outcomes in twin pregnancy: a systematic review and meta-analysis. J Perinatol. 2023 Feb;43(2):133-146. doi: 10.1038/s41372-022-01534-y
Giannubilo SR, Fiorelli A, Marzioni D, Tossetta G, Capogrosso G, Ciavattini A. Maternal Inherited Thrombophilia in Monochorionic Twin Pregnancy with Twin-Twin Transfusion Syndrome. J Clin Med. 2022 Nov 29;11(23):7054. doi: 10.3390/jcm11237054
- As suggested, we have expanded the "Discussion" section of the manuscript, taking into account two of the three recent publications mentioned by the reviewer. Regarding Ref.3, the study focuses on the potential influence of hereditary thrombophilia on the occurrence of TTTS. Since they did not evaluate any of the main outcomes of our study, we were unable to engage in a comparative discussion with our results.
REFERENCES
- Authors are requested to remove the double numbering of the list of references
- In accordance with the reviewer's suggestion, we have revised the list of references.
Reviewer 2 Report
The introduction is well-conceived, but it is necessary to add another paragraph about the types of laser coagulation techniques.
In the materials and methods section, you must clearly state which laser coagulation technique was used in your center, and by how many operators it was used in your series of patients.
You must clearly state in how many cases the placenta was in the back and how many in the front, because we know that access is much more difficult if it is in the front.
When you mention ultrasound and laser coagulation, you must clearly state all information about the devices with which the fetuses were examined and treated in your study.
By how many researchers were the data extracted from the electronic database. Did you calculate the inter rater reliability?
Remove section 2.6. considering that according to the template, the data are listed at the end of the manuscript.
Are the groups comparable by gender?
In relation to previous studies, why didn't you do a power analysis?
Why didn't you take into account more maternal variables that may have an impact on NDI?
How would you rule out a confounder related to different environmental influences on your observed children up to the age of two?
Consider additional limitations of your study that you did not recognize and include them in your study.
In conclusion, also emphasize laser coagulation as a relatively safe method for TTTS.
Emphasize that informed consent was obtained from the parents of the children included in the study.
A moderate correction of English is needed.
Author Response
- The introduction is well-conceived, but it is necessary to add another paragraph about the types of laser coagulation techniques.
- As suggested by the reviewer, we have added a paragraph in the introduction that presents and evaluates the different techniques for performing fetoscopic laser procedures.
- In the materials and methods section, you must clearly state which laser coagulation technique was used in your center, and by how many operators it was used in your series of patients.
- We have incorporated the requested information in the section "2.1. Study design, setting, and population," providing details about the technique employed and the number of operators. This information enhances the transparency of our study design and contributes to the understanding of the expertise involved in the implementation of the fetoscopic interventions.
- You must clearly state in how many cases the placenta was in the back and how many in the front, because we know that access is much more difficult if it is in the front.
- The information requested by the reviewer can be found in Table 2. The table presents the distribution of placental location, with 63.3% of the cases showing a posterior uterine position.
- When you mention ultrasound and laser coagulation, you must clearly state all information about the devices with which the fetuses were examined and treated in your study.
- As suggested, we have included information in the Materials and Methods section regarding the ultrasound equipment and laser used for conducting fetoscopy.
- By how many researchers were the data extracted from the electronic database. Did you calculate the inter rater reliability?
- Data extraction was conducted solely by one of the investigators (A.C.), ensuring consistency and accuracy in the collection process.
- Remove section 2.6. considering that according to the template, the data are listed at the end of the manuscript.
- In response to the suggestion, we have eliminated Section 2.6 from the manuscript due to its redundant nature.
- Are the groups comparable by gender?
- Thank you for the inquiry. Figure 2 illustrates the distribution of fetal sex in TTTS group. Among the TTTS group, 59.5% of the fetuses were female. Similarly, in the Uncomplicated MCDA group, 53.7% of the fetuses (72 out of 134) were female.
- In relation to previous studies, why didn't you do a power analysis?
- In the present study, our aim was to determine the impact of fetoscopic laser procedures on pregnancies affected by TTTS. To achieve this, we conducted a comparative analysis with a sample of uncomplicated monochorionic pregnancies serving as the control group. We assessed differences in terms of NDI at two years of age, calculating the adjusted odds ratio (OR) considering birth weight and gestational age. Similar to the majority of referenced publications, our study was retrospective in nature, the available sample is fixed and determined by the inclusion criteria and the timeframe of data collection. Since the entire population of patients with the pathology of interest is included, there is no need to estimate sample size for adequate statistical power.
It is important to note that although a power analysis may not be necessary in retrospective studies with a complete sample, it is still crucial to acknowledge this limitation and interpret the results accordingly. Future studies with prospective designs can incorporate power analysis to determine appropriate sample sizes and enhance the robustness of statistical analyses.
- Why didn't you take into account more maternal variables that may have an impact on NDI?
- In order to evaluate the impact of fetoscopic laser procedures on neurodevelopment, we conducted this study by comparing it with a cohort of uncomplicated monochorionic pregnancies to establish the baseline risk associated with monochorionicity. Ideally, the optimal control group would have been monochorionic twin pregnancies complicated by TTTS but not subjected to fetoscopic laser intervention. However, ethical considerations based on accumulated evidence prevent us from depriving patients of this treatment, making such a comparison unfeasible.
Furthermore, we aimed to adjust the results by considering three parameters that the literature has consistently linked to a significant impact on neurodevelopment: birth order, birth weight (per gram), and gestational age at birth (per day) [1,2]. Other maternal or fetal parameters significantly associated with neurodevelopment in the first two years of life were not found in the literature.
[1] Lutfi S, Allen VM, Fahey J, O'Connell CM, Vincer MJ. Twin-twin transfusion syndrome: a population-based study. Obstet Gy-necol 2004; 104 : 1289-1297.
[2]. Frusca T, Soregaroli M, Fichera A, Taddei F, Villani P, Accorsi P, Martelli P. Pregnancies complicated by Twin-Twin transfu-sion syndrome: outcome and long-term neurological follow-up. Eur J Obstet Gynecol Reprod Biol 2003; 107 : 145-150.
- How would you rule out a confounder related to different environmental influences on your observed children up to the age of two?
- We appreciate the reviewer's question. While we acknowledge that social and environmental factors can contribute to neurodevelopmental outcomes, we opted not to stratify for these factors based on several considerations.
First, our study focused specifically on the impact of fetal therapy on neurodevelopment, aiming to assess the direct effect of the surgical intervention. By comparing the experimental group (undergoing fetal surgery) with the control group, we aimed to isolate the effect of the surgical procedure itself, assuming that the distribution of social and environmental factors would be similar between the two groups.
Second, conducting a thorough assessment and stratification for social and environmental factors would require comprehensive data collection and extensive resources, which were beyond the scope and resources available for this study. Furthermore, considering that the literature referenced in our study also did not stratify for these factors [1-25], we followed the established precedent.
[1]. De Lia JE, Kuhlmann RS, Lopez KP. Treating previable twin-twin transfusion syndrome with fetoscopic laser surgery: out-comes following the learning curve. J Perinat Med 1999; 27 : 61-67.
[2]. Sutcliffe AG, Sebire NJ, Pigott AJ, Taylor B, Edwards PR, Nicolaides KH. Outcome for children born after in utero laser abla-tion therapy for severe twin-to-twin transfusion syndrome. BJOG 2001; 108 : 1246-1250.
[3]. Banek CS, Hecher K, Hackeloer BJ, Bartmann P. Long-term neurodevelopmental outcome after intrauterine laser treatment for severe twin-twin transfusion syndrome. Am J Obstet Gynecol 2003; 188 : 876-880.
[4]. Graef C, Ellenrieder B, Hecher K, Hackeloer BJ, Huber A, Bartmann P. Long-term neurodevelopmental outcome of 167 chil-dren after intrauterine laser treatment for severe twin-twin transfusion syndrome. Am J Obstet Gynecol 2006; 194 : 303-308.
[5]. Lopriore E, Ortibus E, Acosta-Rojas R, Le Cessie S, Middeldorp JM, Oepkes D, Gratacos E, Vandenbussche FP, Deprest J, Wal-ther FJ, Lewi L. Risk factors for neurodevelopment impairment in twin-twin transfusion syndrome treated with fetoscopic laser surgery. Obstet Gynecol 2009; 113 : 361-366.
[6]. Lenclen R, Ciarlo G, Paupe A, Bussieres L, Ville Y. Neurodevelopmental outcome at 2 years in children born preterm treated by amnioreduction or fetoscopic laser surgery for twin-to-twin transfusion syndrome: comparison with dichorionic twins. Am J Obstet Gynecol 2009; 201 : 291.e1-5.
[7]. Salomon LJ, Ortqvist L, Aegerter P, Bussieres L, Staracci S, Stirnemann JJ, Essaoui M, Bernard JP, Ville Y. Long-term develop-mental follow-up of infants who participated in a randomized clinical trial of amniocentesis vs laser photocoagulation for the treatment of twin-to-twin transfusion syndrome. Am J Obstet Gynecol 2010; 203 : 444.e1-7.
[8]. Sago H, Hayashi S, Saito M, Hasegawa H, Kawamoto H, Kato N, Nanba Y, Ito Y, Takahashi Y, Murotsuki J, Nakata M, Ishii K, Murakoshi T. The outcome and prognostic factors of twin-twin transfusion syndrome following fetoscopic laser surgery. Prenat Diagn 2010; 30: 1185-1191
[9]. Gray PH, Poulsen L, Gilshenan K, Soong B, Cincotta RB, Gardener G. Neurodevelopmental outcome and risk factors for disa-bility for twin-twin transfusion syndrome treated with laser surgery. Am J Obstet Gynecol 2011; 204 : 159.e1-6.
[10]. Chang YL, Chao AS, Chang SD, Lien R, Hsieh PC, Wang CN. The neurological outcomes of surviving twins in severe twin-twin transfusion syndrome treated by fetoscopic laser photocoagulation at a newly established center. Prenat Diagn 2012; 32 : 893-896.
[11]. Graeve P, Banek C, Stegmann-Woessner G, Maschke C, Hecher K, Bartmann P. Neurodevelopmental outcome at 6 years of age after intrauterine laser therapy for twin-twin transfusion syndrome. Acta Paediatr 2012; 101 : 1200-1205.
[12]. Kowitt B, Tucker R, Watson-Smith D, Muratore CS, O'Brien BM, Vohr BR, Carr SR, Luks FI. Long-term morbidity after fetal endoscopic surgery for severe twin-to-twin transfusion syndrome. J Pediatr Surg 2012; 47 : 51-56.
[13]. Swiatkowska-Freund M, Pankrac Z, Preis K. Results of laser therapy in twin-to-twin transfusion syndrome: our experience. J Matern Fetal Neonatal Med 2012; 25 : 1917-1920.
[14]. Tosello B, Blanc J, Haumonté JB, D'Ercole C, Gire C. Short and medium-term outcomes of live-born twins after fetoscopic laser therapy for twin-twin transfusion syndrome. J Perinat Med 2014; 42 : 99-105.
[15]. McIntosh J, Meriki N, Joshi A, Biggs V, Welsh AW, Challis D, Lui K. Long term developmental outcomes of pre-school age children following laser surgery for twin-to-twin transfusion syndrome. Early Hum Dev 2014; 90 : 837-842.
[16]. Vanderbilt DL, Schrager SM, Llanes A, Hamilton A, Seri I, Chmait RH. Predictors of 2-year cognitive performance after laser surgery for twin-twin transfusion syndrome. Am J Obstet Gynecol 2014; 211 : 388.e1-7.
[17]. Müllers SM, McAuliffe FM, Kent E, Carroll S, Mone F, Breslin N, Dalrymple J, Mulcahy C, O'Donoghue K, Martin A, Malone FD. Outcome following selective fetoscopic laser ablation for twin to twin transfusion syndrome: an 8 year national collaborative experience. Eur J Obstet Gynecol Reprod Biol 2015; 191 : 125-129.
[18]. van Klink JM, Slaghekke F, Balestriero MA, Scelsa B, Introvini P, Rustico M, Faiola S, Rijken M, Koopman HM, Middeldorp JM, Oepkes D, Lopriore E. Neurodevelopmental outcome at 2 years in twin-twin transfusion syndrome survivors randomized for the Solomon trial. Am J Obstet Gynecol 2016; 214 : 113.e1-7.
[19]. Sananès N, Gabriele V, Weingertner AS, Ruano R, Sanz-Cortes M, Gaudineau A, Langer B, Nisand I, Akladios CY, Favre R. Evaluation of long-term neurodevelopment in twin-twin transfusion syndrome after laser therapy. Prenat Diagn 2016; 36 : 1139-1145.
[20]. Campos D, Arias AV, Campos-Zanelli TM, Souza DS, Dos Santos Neto OG, Peralta CF, Guerreiro MM. Twin-twin transfusion syndrome: neurodevelopment of infants treated with laser surgery. Arq Neuropsiquiatr 2016; 74 : 307-313.
[21]. Korsakissok M, Groussolles M, Dicky O, Alberge C, Casper C, Azogui-Assouline C. Mortality, morbidity and 2-years neuro-developmental prognosis of twin to twin transfusion syndrome after fetoscopic laser therapy: a prospective, 58 patients cohort study. J Gynecol Obstet Hum Reprod 2018; 47 : 555-560.
[22]. Sommer J, Nuyt AM, Audibert F, Dorval V, Wavrant S, Altit G, Lapointe A. Outcomes of extremely premature infants with twin-twin transfusion syndrome treated by laser therapy. J Perinatol 2018; 38 : 1548-1555.
[23]. Schou KV, Lando AV, Ekelund CK, Jensen LN, Jørgensen C, Nørgaard LN, Rode L, Søgaard K, Tabor A, Sundberg K. Long-Term Neurodevelopmental Outcome of Monochorionic Twins after Laser Therapy or Umbilical Cord Occlusion for Twin-Twin Transfusion Syndrome. Fetal Diagn Ther 2019; 46 : 20-27.
[24]. Matsushima S, Ozawa K, Sugibayashi R, Ogawa K, Tsukamoto K, Miyazaki O, Wada S, Ito Y, Sago H. Neurodevelopmental impairment at 3 years of age after fetoscopic laser surgery for twin-to-twin transfusion syndrome. Prenat Diagn 2020; 40 : 1013-1019.
[25]. Chimenea A, García-Díaz L, Antiñolo G. Mode of delivery, perinatal outcome and neurodevelopment in uncomplicated mon-ochorionic diamniotic twins: a single-center retrospective cohort study. BMC Pregnancy Childbirth 2022; 22 : 89.
- Consider additional limitations of your study that you did not recognize and include them in your study.
- As requested by the reviewer, we have expanded on the limitations associated with our study.
- In conclusion, also emphasize laser coagulation as a relatively safe method for TTTS.
- As requested by the reviewer, we have included the requested information in the conclusion
- Emphasize that informed consent was obtained from the parents of the children included in the study.
- As requested by the reviewer, we have included the requested information in the Informed Consent Statement.
Round 2
Reviewer 2 Report
I thank you for the answers and explanations, which improved your own manuscript.
Minor editing required.